# Learning Representations from Audio-Visual Spatial Alignment

**Pedro Morgado**[*]     **Yi Li**[*]     **Nuno Vasconcelos**

Department of Electrical and Computer Engineering
University of California, San Diego
{pmaravil,yil898,nuno}@eng.ucsd.edu

## Abstract

We introduce a novel self-supervised pretext task for learning representations from audio-visual content. Prior work on audio-visual representation learning leverages correspondences at the video level. Approaches based on audio-visual correspondence (AVC) predict whether audio and video clips originate from the same or different video instances. Audio-visual temporal synchronization (AVTS) further discriminates negative pairs originated from the same video instance but at different moments in time. While these approaches learn high-quality representations for downstream tasks such as action recognition, their training objectives disregard spatial cues naturally occurring in audio and visual signals. To learn from these spatial cues, we tasked a network to perform contrastive audio-visual spatial alignment of $360°$ video and spatial audio. The ability to perform spatial alignment is enhanced by reasoning over the full spatial content of the $360°$ video using a transformer architecture to combine representations from multiple viewpoints. The advantages of the proposed pretext task are demonstrated on a variety of audio and visual downstream tasks, including audio-visual correspondence, spatial alignment, action recognition and video semantic segmentation. Dataset and code are available at https://github.com/pedro-morgado/AVSpatialAlignment.

## 1   Introduction

Human perception is inherently multi-sensory. Since real-world events can manifest through multiple modalities, the ability to integrate information from various sensory inputs can significantly benefit perception. In particular, neural processes for audio and visual perception are known to influence each other significantly. These interactions are responsible for several well known audio-visual illusions such as the "McGurk effect" [38], the "sound induced flash effect" [52] or the "fusion effect" [2], and can even be observed in brain activation studies, where areas of the brain dedicated to visual processing have been shown to be activated by sounds that are predictive of visual events, even in the absence of visual input [14, 58].

In computer vision, the natural co-occurrence of audio and video has been extensively studied. Prior work has shown that this co-occurrence can be leveraged to learn representations in a self-supervised manner, i.e., without human annotations. A common approach is to learn to match audio and video clips of the same video instance [3, 4, 41]. Intuitively, if visual events are associated with a salient sound signature, then the audio can be treated as a label to describe the visual content [49]. Prior work has also demonstrated the value of temporal synchronization between audio and video clips for learning representations for downstream tasks such as action recognition [30, 46].

---

[*]Equal contribution.

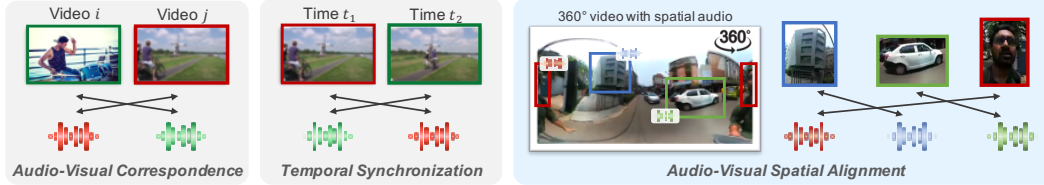

**Figure 1: Audio-visual spatial alignment.** Prior work on audio-visual representation learning leverages correspondences at the video level. Audio-visual correspondence (AVC) [3, 4, 41] predicts whether a pair of audio and video clips originate from the same video (positive) or different videos (negative). Audio-visual temporal synchronization (AVTS) [30, 46] discriminates negative pairs that are sampled from the same video but different moments in time. However, prior work ignores the spatial cues of audio-visual signals. Instead, we learn representations by performing audio-visual spatial alignment (AVSA) of 360°video and spatial audio. This is accomplished by training a model to distinguish audio and video clips extracted from different viewpoints.

Since these methods do not need to localize sound sources, they struggle to discriminate visual concepts that often co-occur. For example, the sound of a car can be quite distinctive, and thus it is a good target description for the "car" visual concept. However, current approaches use this audio as a descriptor for the whole video clip, as opposed to the region containing the car. Since cars and roads often co-occur, there is an inherent ambiguity about which of the two produce the sound. This makes it is hard to learn good representations for visual concepts like "cars", distinguishable from co-occurring objects like "roads" by pure audio-visual correspondence or temporal synchronization. This problem was clearly demonstrated in [51] that shows the poor audio localization achieved with AVC pretext training.

To address this issue, we learn representations by training deep neural networks with 1) 360°video data that contain audio-visual signals with strong spatial cues and 2) a pretext task to conduct audio-visual spatial alignment (AVSA, Figure 1). Unlike regular videos with mono audio recordings, 360°video data and spatial audio formats like ambisonics fully capture the spatial layout of audio and visual content within a scene. To learn from this spatial information, we collected a large 360°video dataset, five times larger than currently available datasets. We also designed a pretext task where audio and video clips are sampled from different viewpoints within a 360°video, and spatially misaligned audio/video clips are treated as negatives examples for contrastive learning. To enhance the learned representations, two modifications to the standard contrastive learning setup are proposed. First, the ability to perform spatial alignment is boosted using a curriculum learning strategy that initially focus on learning audio-visual correspondences at the video level. Second, we propose to reason over the full spatial content of the 360°video by combining representations from multiple viewpoints using a transformer network. We show the benefits of the AVSA pretext task on a variety of audio and visual downstream tasks, including audio-visual correspondence and spatial alignment, action recognition and video semantic segmentation.

## 2 Related work

**360°media** The increasing availability of 360°data has sparked interest in developing vision systems for 360°imagery. For example, the SUN-360 dataset of static 360°images was collected to learn to recognize viewpoints within a scene [63]. Self-supervised monocular depth and camera motion estimation have also been studied by pairing 360°imagery with depth data [33, 60]. Another common topic of interest is to enhance 360°video consumption by guiding the viewer towards salient viewing angles within a video [10, 65], automating the field-of-view control for 360°video playback [24, 54], or by upgrading mono recordings into spatial sounds [40].

**Self-supervised learning** Self-supervised learning methods learn representations without requiring explicit human annotation. Instead of predicting human labels, self-supervision learns representations that are predictive of the input data itself (or parts of it) while imposing additional constraints such as sparsity [34, 43, 44] or invariance [8, 20, 25, 39, 48]. An emergent technique, known as contrastive learning, relies on contrastive losses [20] to learn view invariant representations, where the different views of the data can be generated by data augmentation [9, 23, 39, 62], chunking the input over time or space [21, 45] or using co-occurring modalities [3, 26, 41, 55, 64]. In this work, we also rely on contrastive losses, but utilize contrastive learning to perform audio-visual spatial alignment.

Similarly to the proposed AVSA task, spatial context has previously been used in visual representation learning. For example, [12, 27, 42] try to predict the relative locations of image or video patches, and [21] uses contrastive learning to learn representations that are predictive of their spatio-temporal location. However, as shown in [41], using visual content as both the input and target for representation learning can yield sub-optimal representations, as low-level statistics can be explored to perform the task without learning semantic features. Our approach addresses this issue by leveraging the spatial context provided by a co-occurring modality (audio) that also contains strong spatial cues.

**Audio-visual learning**    The natural co-occurrence of vision and sound has been successfully used in various contexts such as visually guided source separation and localization [15, 16, 18, 66, 67], and audio spatialization [17, 40]. Audio-visual correspondences [3, 4] have also been used for learning representations for objects and scenes in static images [3, 4, 47], action recognition in video [1, 30, 41, 46], to perform temporal synchronization [11, 22, 30, 46] and audio classification [5]. As discussed in Figure 1, prior work is often implemented either by predicting audio-visual correspondences at the video level [3, 4, 41] or performing temporal synchronization using out-of-sync clips as hard negatives [30, 46]. However, [51] shows that basic audio-visual correspondences are ill-equipped to identify and localize sound sources in the video. We argue that this is because audio-visual correspondences are imposed by matching audio to the entire video clip. Thus, there is little incentive to learn discriminative features for objects that often co-occur. To address this issue, we explore the rich spatial cues present in both the 360°video and spatial audio. By learning to spatially align visual and audio contents, the network is encouraged to reason about the scene composition (i.e. the locations of the various sources of sound), thus yielding better representations for downstream tasks.

## 3    Audio-visual spatial alignment

We learn audio-visual representations by leveraging spatial cues in 360°media. 360°video and spatial audio encode visual and audio signals arriving from all directions $(\theta, \phi)$ around the recording location, where $\theta$ denotes the longitude (or horizontal) angle, $\phi$ the latitude (or elevation) angle. We adopt the equi-rectangular projection as the 360°video format and first-order ambisonics [19] for the spatial audio. Both formats can be easily rotated and/or decoded into viewpoint specific clips.

### 3.1    Pretext task

**Regressive AVSA**    A straight-forward implementation of audio-visual spatial alignment is to generate random rotations $R$ of either the video or audio so as to create an artificial misalignment between them. A model can then be trained to predict the applied transformation by solving

$$\min_{f_v, f_a, g} \mathbb{E}_{v,a,R} \left\{ d\left[ g(f_v(v), f_a(R(a))), R \right] \right\}, \tag{1}$$

where $f_v$ and $f_a$ are the video and audio encoders, $g$ a rotation regression head, and $d$ the distance between the predicted and ground-truth rotations $R$. However, this implementation has several disadvantages. Due to the continuous nature of the target variable $R$, the loss of (1) is difficult to optimize. Also, the task is defined on the full 360°video $v$, which limits the use of data augmentation techniques such as aggressive cropping that are critical for self-supervised learning.

**Contrastive AVSA**    Inspired by recent advances in contrastive learning [20, 41, 45, 55, 62], we propose to solve the audio-visual spatial alignment task in a contrastive fashion. As shown in Figure 1, given a 360°audio-video sample $(v_i, a_i)$, $K$ video and audio clips $\{(v_i^k, a_i^k)\}_{k=1}^K$ are extracted from $K$ randomly sampled viewing directions $\{(\theta_k, \phi_k)\}_{k=1}^K$. Video clips $v_i^k$ are obtained by extracting normal field-of-view (NFOV) crops using a Gnomonic projection [61] centered around $(\theta_k, \phi_k)$, and audio clips $a_i^k$ by realigning the global frame of reference of the ambisonics signal such that the frontal direction points towards $(\theta_k, \phi_k)$ [31]. Audio-visual spatial alignment is then encouraged by tasking a network to predict the correct correspondence between the $K$ video $\{v_i^k\}_{k=1}^K$ and the $K$ audio $\{a_i^k\}_{k=1}^K$ signals.

### 3.2    Architecture

Figure 2 summarizes the architecture used to solve the spatial alignment task. First, video and audio encoders, $f_v$ and $f_a$, extract feature representations from each clip independently,

$$\mathbf{v}_i^k = f_v(v_i^k) \quad \text{and} \quad \mathbf{a}_i^k = f_a(a_i^k). \tag{2}$$

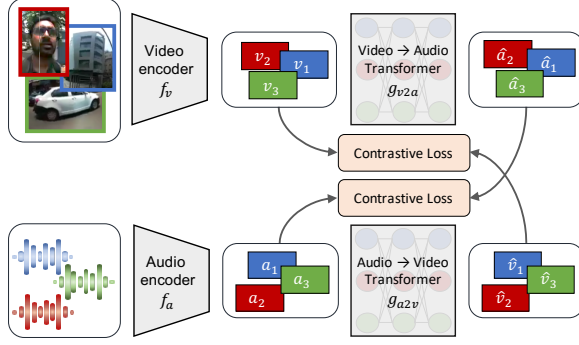

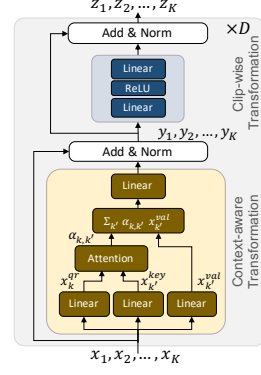

**Figure 2:** Architecture overview for contrastive audio-visual spatial alignment.

**Figure 3:** Transformer architecture for context-aware video-to-audio and audio-to-video feature translation.

These representations are then converted between the two modalities using audio-to-video $g_{a2v}$ and video-to-audio $g_{v2a}$ feature translation networks

$$\bar{\mathbf{v}}_i^1, \ldots, \bar{\mathbf{v}}_i^K = g_{a2v}(\mathbf{a}_i^1, \ldots, \mathbf{a}_i^K) \quad \text{and} \quad \bar{\mathbf{a}}_i^1, \ldots, \bar{\mathbf{a}}_i^K = g_{v2a}(\mathbf{v}_i^1, \ldots, \mathbf{v}_i^K). \tag{3}$$

One important distinction between audio and video is the spatial localization of the signals. Unlike video, any sound source can be heard regardless of the listening angle. In other words, while an audio clip $a_i^k$ sampled at position $(\theta_k, \phi_k)$ contains audio from all sound sources present in a scene, only those physically located around $(\theta_k, \phi_k)$ can be seen on the video clip $v_i^k$. This implies that, to enable accurate feature translation, networks $g_{v2a}$ and $g_{a2v}$ should combine features from all sampled locations. This is accomplished by using a translation network similar to the transformer of [57]. As shown in Fig. 3, given a set of $K$ features $\{\mathbf{x}_k\}_{k=1}^K$, a transformer of depth $D$ alternates $D$ times between two modules. The first module combines the $K$ features $\mathbf{x}_k$ using attention

$$\alpha_{k,1}, \ldots, \alpha_{k,K} \quad = \quad \textit{Softmax}\left(\frac{\langle W_{key}^T \mathbf{x}_k, W_{qr}^T \mathbf{x}_1 \rangle}{\sqrt{d}}, \ldots, \frac{\langle W_{key}^T \mathbf{x}_k, W_{qr}^T \mathbf{x}_K \rangle}{\sqrt{d}}\right) \tag{4}$$

$$\mathbf{y}_k \quad = \quad \textit{Norm}\left(\mathbf{x}_k + W_0^T \sum_{k'} \alpha_{k,k'} W_{val}^T \mathbf{x}_{k'}\right). \tag{5}$$

The second module computes a simple clip-wise feed-forward transformation

$$\mathbf{z}_k \quad = \quad \textit{Norm}\left(\mathbf{y}_k + W_2^T \max(W_1^T \mathbf{y}_k, 0)\right). \tag{6}$$

In (4)-(6), $W_{key}, W_{qr}, W_{val}, W_0, W_1$ and $W_2$ are learnable weights and *Norm* is layer normalization [6]. We omit the biases of linear transformations and layer indices for simplicity of notation. Compared to the original transformer [57], the proposed translation network differs in two aspects. First, motivated by early empirical results which showed no improvements on downstream tasks when utilizing multi-head attention, we simplified the transformer architecture to rely on a single attention head. Second, we removed positional encodings which are used to indicate the position of each token $\mathbf{x}_k$. While these encodings could be used to encode the viewing direction $(\theta_k, \phi_k)$ of each clip, doing so would allow the model to solve the spatial alignment task without learning semantic representations.

### 3.3 Learning strategy

AVSA is a difficult task to optimize since it requires discriminating between various crops from the same video. To enhance learning, we employed a curriculum learning strategy [7]. In the first phase, the network is trained to identify audio-visual correspondences (AVC) [3, 41] at the video level. This is accomplished by extracting a single crop $(v_i, a_i)$ for each video $i$ from a randomly drawn viewing angle. The visual and audio encoders, $f_v$ and $f_a$, are then trained to minimize

$$L_{AVC} = \sum_i L_{\textit{InfoNCE}}\left(\mathbf{v}_i, \mathbf{a}_i, \{\mathbf{a}_j\}_{j=1}^N\right) + L_{\textit{InfoNCE}}\left(\mathbf{a}_i, \mathbf{v}_i, \{\mathbf{v}_j\}_{j=1}^N\right) \tag{7}$$

| | Spatial Audio | Unique Videos | Hours |
|---|---|---|---|
| Duanmu *et al.* [13] | | 12 | 0.3 |
| Li *et al.* [35] | | 73 | 3.8 |
| Pano2VID [54] | | 86 | 7.3 |
| SptAudioGen [40] | ✓ | 1146 | 113 |
| YT-360 | ✓ | 5506 | 246 |

**Table 1:** Comparison of 360° video datasets.

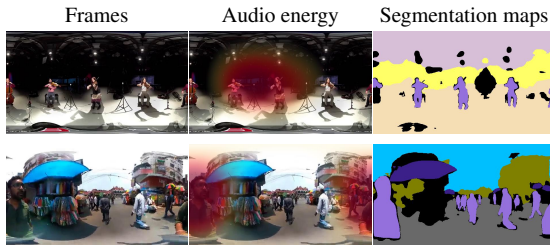

Frames   Audio energy   Segmentation maps

**Figure 4:** Examples from Youtube-360 dataset.

where $\mathbf{v}_i = f_v(v_i)$ and $\mathbf{a}_i = f_a(a_i)$ are the video and audio representations. $L_{InfoNCE}$ is the InfoNCE loss [45] defined as

$$L_{InfoNCE}(\mathbf{x}, \mathbf{x}_t, \mathcal{P}_\mathbf{x}) = -\log \frac{\exp(h(\mathbf{x}_t, \mathbf{x})/\tau)}{\sum_{\mathbf{x}_p \in \mathcal{P}_\mathbf{x}} \exp(h(\mathbf{x}_p, \mathbf{x})/\tau)} \tag{8}$$

where $h(\mathbf{x}, \mathbf{x}_t)$ is a prediction head that computes the cosine similarity between $\mathbf{x}$ and $\mathbf{x}_t$ after linear projection into a low-dimensional space, and $\tau$ is a temperature hyper-parameter. In the case of AVC, the target representation $\mathbf{x}_t$ for the InfoNCE loss is the feature from the crop of same video but opposing modality, and the proposal distribution $\mathcal{P}_\mathbf{x}$ is composed by the target feature representations of all videos in the batch.

In the second phase, the network is trained on the more challenging task of matching audio and video at the crop level, i.e. matching representations in the presence of multiple crops per video. This is accomplished by augmenting the proposal set $\mathcal{P}_\mathbf{x}$ to include representations from multiple randomly sampled viewing angles $\{(v_i^k, a_i^k)\}_{k=1}^K$ from the same video. In this phase, we also introduce the feature translation networks $g_{v2a}$ and $g_{a2v}$ and require the translated features ($\bar{\mathbf{v}}_i^k$ and $\bar{\mathbf{a}}_i^k$) to match the encoder outputs ($\mathbf{v}_i^k$ and $\mathbf{a}_i^k$) obtained for the corresponding viewing angle $k$. Encoders $f_v$ and $f_a$ and feature translation networks $g_{v2a}$ and $g_{a2v}$ are jointly trained to minimize

$$L_{AVSA} = \sum_i \sum_k L_{InfoNCE}\left(\bar{\mathbf{v}}_i^k, \mathbf{v}_i^k, \{\mathbf{v}_j^l\}_{j,l=1}^{N,K}\right) + L_{InfoNCE}\left(\bar{\mathbf{a}}_i^k, \mathbf{a}_i^k, \{\mathbf{a}_j^l\}_{j,l=1}^{N,K}\right). \tag{9}$$

## 4  YouTube-360 dataset

We collected a dataset of 360° video with spatial audio from YouTube, containing clips from a diverse set of topics such as musical performances, vlogs, sports, and others. This diversity is critical to learn good representations. Similarly to prior work [40], search results were cleaned by removing videos that 1) did not contain valid ambisonics, 2) only contain still images, or 3) contain a significant amount of post-production sounds such as voice-overs and background music. The resulting dataset, denoted YouTube-360 (YT-360), contains a total of 5 506 videos, which was split into 4 506 videos for training and 1 000 for testing. Since we use audio as target for representation learning, periods of silence were ignored. This was accomplished by extracting short non-overlapping clips whose volume level is above a certain threshold. In total, 88 733 clips of roughly 10s each were collected (246 hours of video content). As shown in Table 1, the YT-360 dataset contains five times more videos than the largest 360° video dataset previously collected.

To assess the ability of AVSA pre-training to localize objects in a scene, we conduct evaluations on semantic segmentation as a downstream task. Due to the large size of our dataset, collecting ground-truth annotations is impractical. Instead, we used the state-of-the-art ResNet101 Panoptic FPN model [29] trained on the MS-COCO dataset [37] to segment the 32 most frequent objects and background classes on YT-360. A description of the segmentation procedure, including the selected classes, is provided in appendix. These segmentation maps are used to evaluate AVSA representations by knowledge distillation, as discussed in Section 5.3. Examples from the YT-360 dataset are shown in Figure 4 together with the predicted segmentation maps and a heat-map representing the directions of higher audio volume.

# 5 Experiments

We evaluate the representations learned by AVSA pre-training on several downstream tasks. We explain the experimental setting below, and refer the reader to appendix for additional details.

## 5.1 Experimental setting

**Video pre-processing**  We sampled $K = 4$ crops per video at different viewing angles. Since up and down viewing directions are often less informative, we restrict the center of each crop to latitudes $\phi \in \{-60°, 60°\}$. We also ensure that viewing angles are sampled at least 36° apart. Normal field-of-view (NFOV) crops are extracted using a Gnomonic projection with random angular coverage between 25° and 90° wide for data augmentation. If naive equi-rectangular crops were taken, the distortion patterns of these crops at latitudes outside the horizon line could potentially reveal the vertical position of the crop, allowing the network to "cheat" the AVSA task. Following NFOV projection, video clips are resized into $112 \times 112$ resolution. Random horizontal flipping, color jittering and Z normalization are applied. Each video clip is $0.5s$ long and is extracted at 16fps.

**Audio pre-processing**  First-order ambisonics (FOA) are used for spatial audio. Audio clips for the different viewing angles are generated by simply rotating the ambisonics [31]. One second of audio is extracted at 24kHz, and four channels (FOA) of normalized log mel-spectrograms are used as the input to the audio encoder. Spectrograms are computed using an STFT with a window of size 21ms, and hop size of 10ms. The extracted frequency components are aggregated in a mel-scale with 128 levels.

**Architecture and optimization**  The video encoder $f_v$ is the 18-layer R2+1D model [56], and the audio encoder $f_a$ is a 9-layer 2D convolutional neural network operating on the time-frequency domain. The translation networks, $g_{v2a}$ and $g_{a2v}$, are instantiated with depth $D = 2$. Training is conducted using the Adam optimizer [28] with a batch size of 28 distributed over 2 GPUs, learning rate of $1e-4$, weight decay of $1e-5$ and default momentum parameters $(\beta_1, \beta_2) = (0.9, 0.999)$. Both curriculum learning phases are trained for 50 epochs. To control for the number of iterations, models trained only on the first or second phases are trained for 100 epochs.

**Baseline pre-training methods**  We compare AVSA to Audio-Visual Correspondence (AVC) [3, 4, 41] and Audio-Visual Temporal Synchronization (AVTS) [30, 46]. Since prior works perform pretext training on flat video datasets (i.e. without spatial audio), a direct comparison is impossible. Instead, we train AVC and AVTS models on the YouTube-360 dataset. For fair comparisons, we use the architecture and optimization settings described above. AVC is trained to optimize the loss of (7), which only uses negatives from different videos. Note that (7) is similar to the loss used in [3, 4] but considers multiple negatives simultaneously. This has actually been shown to improve generalization in [41]. To implement AVTS, we augment the proposal set $\mathcal{P}_\mathbf{x}$ of the InfoNCE loss of (8) with clips sampled from different moments in time. Following [30, 46], we ensure that negative pairs of audio and video clips are sufficiently separated in time. We also use a curriculum learning strategy composed by an AVC pre-training phase as in [30]. In the base AVC and AVTS implementations, we directly match the audio and visual features computed by the encoders $f_v$ and $f_a$ directly, as done in the original papers [3, 30, 41, 46]. However, to control for the number of seen crops, we also conduct AVC and AVTS pre-training using multiple crops of the same video and the feature translation networks $g_{a2v}$ and $g_{v2a}$. Since AVC requires predictions at the video level (not for each individual clip), clip representations are combined by max-pooling.

## 5.2 Audio-visual spatial alignment

We start by considering the performance on the AVC and AVSA tasks themselves. AVC performance is measured by randomly generating 50% of audio-video pairs from the same sample (positives), and 50% of pairs from different samples (negatives). Similarly, we designed a binary AVSA evaluation task in which positive audio-video pairs are spatially aligned, while negative pairs were artificially misaligned by randomly rotating the ambisonic audio of a positive pair. Rotations are constrained around the yaw axis (horizontal) to ensure the audio from positive and negative pairs have the same distribution, and thus making the AVSA task more challenging. Since models trained by AVC are not tuned for AVSA evaluation and vice-versa, the pretext models cannot be directly evaluated on the above binary tasks. Instead, we trained a new binary classification head on top of video and audio

features, while keeping pretext representations frozen. Also, since NFOV video crops only cover a small portion of the 360°frame, we also consider predictions obtained by averaging over four viewpoints.

Table 2 shows that the proposed AVSA pretext training mechanism significantly outperforms AVC and AVTS on both evaluation tasks. Remarkably, even though AVC pretext training optimizes for the AVC task directly, representations learned with AVSA outperformed those learned with AVC by more than 6% on the AVC task itself (AVC-Bin). Furthermore, both AVC

| Evaluation Task | AVC-Bin | | AVSA-Bin | |
|---|---|---|---|---|
| # Viewpoints | 1 | 4 | 1 | 4 |
| AVC no transf. | 79.82 | 82.68 | 59.48 | 59.25 |
| transf. | – | 83.87 | – | 61.20 |
| AVTS no transf. | 80.08 | 82.77 | 59.78 | 60.37 |
| transf. | – | 83.77 | – | 60.73 |
| AVSA no transf. | 86.19 | 91.67 | 64.97 | 68.87 |
| transf. | – | 89.83 | – | 69.97 |

**Table 2:** Accuracy of binary AVC and AVSA predictions using one or four viewpoints on the YT-360 test set.

and AVTS models learned by audio-video correspondence or temporal synchronization do not transfer well to the spatial alignment task (AVSA-Bin). In result, AVSA outperforms AVC and AVTS by more than 5% on spatial alignment. By learning representations that are discriminative of different viewpoints, AVSA also learns a more diverse set of features. This is especially helpful when combining information from multiple viewpoints, as demonstrated by the differences in the gains obtained by 4 crop predictions. For example, AVC and AVTS only benefit by a 2-3% gain from 4 crop predictions on the AVC-Bin task, while AVSA performance improves by 5.5%. On the AVSA-Bin task, 4 crop predictions do not improve AVC or AVTS significantly, while AVSA performance still improves by 4%. We also observe improvements by using the transformer architecture in 5 out of 6 configurations (3 pretext tasks × 2 evaluations), showing its effectiveness at combining information from different viewpoints.

### 5.3 Semantic segmentation by knowledge distillation

AVSA representations are also evaluated on semantic segmentation. As shown in Figure 5, the video encoder $f_v$ was used to extract features at multiple scales, which were combined using a feature pyramid network (FPN) [36] for semantic segmentation. To measure the value added by audio inputs, we concatenate the features from the audio encoder $f_a$ at the start of the top-down pathway of the FPN head. Similarly, to measure the benefits of combining features from multiple viewpoints, we concatenate the context-aware representations computed by the feature translation modules $g_{v2a}$ and $g_{a2v}$. Since the goal is to evaluate the pretext representations, networks trained on the pretext task were kept frozen. The FPN head was trained by knowledge distillation, i.e. using the predictions of a state-of-the-art model as targets. We also compare to a fully supervised video encoder pre-trained on Kinetics for the task of action recognition. Similar to the self-supervised models, the fully supervised model was kept frozen. To provide an upper bound on the expected performance, we trained the whole system end-to-end (encoders, feature translation modules and the FPN head). A complete description of the FPN segmentation head and training procedure is given in appendix.

Table 3 shows the pixel accuracy and mean IoU scores obtained using video features alone, or their combination with audio and context features. Examples of segmentation maps obtained with the AVSA model with context features are also shown in Figure 6. The results support several observations. AVSA learns significantly better visual features for semantic segmentation than AVC. This is likely due to the fine-grained nature of the AVSA task which requires discrimination of multiple crops within the same video frame. As a result, AVSA improves the most upon AVC on background classes such as rocks (34.7% accuracy vs. 27.7%), window (46.0% vs. 41.2%), pavement (36.8% vs. 33.3%), sand (42.1% vs. 38.8%), sea (50.1% vs. 46.8%) and road (47.1% vs. 45.1%).

AVSA also learns slightly better visual features than AVTS. While the gains over AVTS using visual features alone are smaller, AVTS cannot leverage the larger spatial context of 360°video data. When context features from four viewpoints are combined, using the translation networks $g_{v2a}$ and $g_{a2v}$, further improvements are obtained. With context features, AVSA yields a 3% mIoU improvement over AVC and 1% over AVTS.

Finally, we evaluated two ablations of AVSA. To verify the benefits of curriculum learning, we optimized the AVSA loss of (9) directly. Without curriculum, AVSA achieved 1.5% worse mIoU (see Table 3 AVSA no curr.). We next verified the benefits of modeling spatial context by disabling the transformer ability to combine information from all viewpoints. This was accomplished by replacing

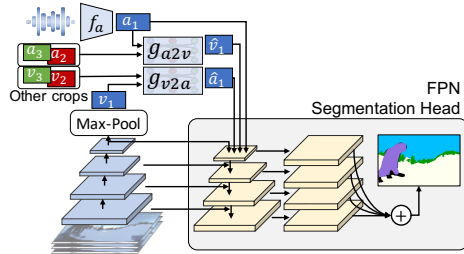

**Figure 5:** Architecture used for semantic segmentation. Pre-trained networks are kept frozen. A lightweight FPN segmentation head [36] is trained by knowledge distillation.

|  | Video only | | +Audio | | +Audio+Context | |
|---|---|---|---|---|---|---|
|  | Pix Acc | mIoU | Pix Acc | mIoU | Pix Acc | mIoU |
| AVC | 71.16 | 32.85 | 71.07 | 32.69 | – | – |
| AVTS | 73.24 | 34.88 | 72.97 | **34.88** | – | – |
| AVSA | **73.44** | **35.11** | **73.11** | 34.63 | **73.85** | **35.83** |
| AVSA (no curr.) | 71.95 | 33.66 | 71.49 | 33.23 | 72.06 | 34.30 |
| AVSA (mlp) | 73.10 | 35.02 | 73.21 | 34.83 | 72.68 | 34.35 |
| Kinetics (sup) | 75.47 | 36.91 | – | – | – | – |
| End-to-end (upper bound) | 77.37 | 41.05 | 77.93 | 42.00 | 79.65 | 43.21 |

**Table 3:** Pixel accuracy and mean IoU of semantic segmentation predictions on YT-360 test set. We evaluate the performance of an FPN head that uses 1) visual features alone, 2) visual and audio features, and 3) visual, audio and context features obtained from four viewpoints.

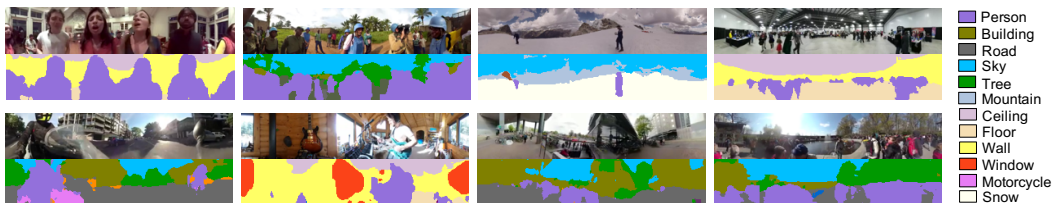

Person
Building
Road
Sky
Tree
Mountain
Ceiling
Floor
Wall
Window
Motorcycle
Snow

**Figure 6:** Predictions from an AVSA pre-trained model with an FPN segmentation head on the YT-360 test set.

the attention module of Figure 3 with a similarly sized multi-layer perceptron, which forced the translation networks to process each viewpoint independently. While this only produced slightly worse visual representations, the ability to leverage spatial context was significantly affected. Without the transformer architecture, AVSA yielded 1.5% worse mIoU scores when using context features (see Table 3 AVSA mlp).

## 5.4 Action recognition

Action recognition is a common downstream task used to benchmark audio-visual self-supervised approaches. Following standard practices, we finetuned the pretext models either on the UCF [53] or the HMDB [32] datasets, and measure the top-1 accuracies obtained for a single clip or by averaging predictions over 25 clips per video. For comparison, we also provide the performance of our model trained on UCF and HMDB from a random initialization (Scratch), or finetuned from a fully supervised model trained on Kinetics [59] (Kinetics Sup.).

|  | UCF | | HMDB | |
|---|---|---|---|---|
|  | Clip@1 | Video@1 | Clip@1 | Video@1 |
| Scratch | 54.85 | 59.95 | 27.40 | 31.10 |
| Kinetics Sup. | 78.50 | 83.43 | 46.45 | 51.90 |
| AVC | 64.63 | 69.68 | 31.33 | 34.58 |
| AVTS | 65.65 | 70.34 | 32.29 | 35.89 |
| AVSA | **68.52** | **73.80** | **32.96** | **37.66** |

**Table 4:** Action recognition performance on UCF and HMDB datasets. The top-1 accuracy of single clip and dense predictions are reported.

Full details of the training procedure are given in appendix. The results shown in Table 4 show once more the benefits of AVSA pretext training. AVSA dense predictions outperform AVC by 4% on UCF and 3% on HMDB, and outperform AVTS by 3.5% on UCF and 2% on HMDB.

## 6 Discussion, future work and limitations

We presented a novel self-supervised learning mechanism that leverages the spatial cues in audio and visual signals naturally occurring in the real world. Specifically, we collected a 360° video dataset with spatial audio, and trained a model to spatially align video and audio clips extracted from different viewing angles. The proposed AVSA task was shown to yield better representations than prior work on audio-visual self-supervision for downstream tasks like audio-visual correspondence, video semantic segmentation, and action recognition. We also proposed to model 360° video data as a collection of NFOV clips collected from multiple viewpoints, using a transformer architecture to

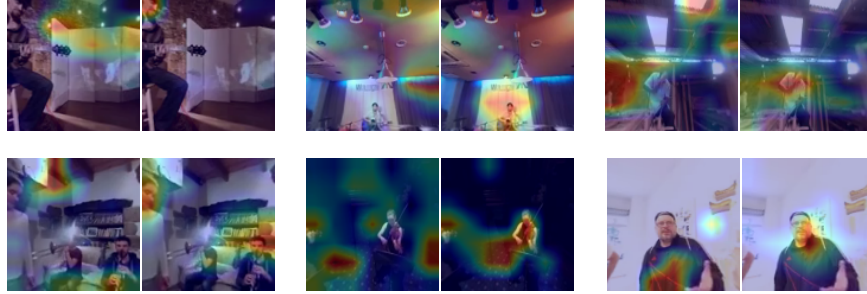

**Figure 7:** Sound localization maps (GradCAM of audio-visual matching scores) obtained from models trained by AVC (first image of each pair) and AVSA (second of each pair).

combine view specific information. Being able to summarize information from the whole 360°video frame was proven advantageous for downstream tasks defined on 360°video data. For additional parametric and ablation studies, we refer the reader to supplementary material, where we ablate several components of the proposed approach, including the type of audio input provided to the network, the number and type of viewpoints in the AVSA objective, and the influence of curriculum learning and the transformer module.

Since AVSA requires discrimination of different viewpoints within a 360°scene, the learned models are encouraged to localize sound sources in the video and audio signals in order to match them. In addition to better performance on downstream tasks, this pre-training objective also translates into improved localization ability, based on a qualitative analysis. Fig. 7 shows several examples of GradCAM [50] visualizations for AVC and AVSA models (GradCAM is applied to each model's audio-visual matching score). As can be seen, AVSA models tend to localize sound sources better. Furthermore, while the proposed method relies on randomly extracted video and audio clips, more sophisticated sampling techniques are an interesting direction of future work. For example, sampling can be guided towards objects using objectness scores, towards moving objects using optical flow, or towards sound sources by oversampling viewpoints with high audio energy. Such sampling techniques would better mimic a human learner, by actively choosing which parts of the environment to dedicate more attention. They would also under-sample less informative viewpoints (e.g. crops dominated by background), which are hard to match to the corresponding sound, and thus may harm the quality of learned representations.

Finally, we note that AVSA requires 360°data with spatial audio, which is still less prevalent than regular video. Previous methods, such as AVC and AVTS [3, 30, 41, 46], are often trained on datasets several orders of magnitude larger than YT-360, and can achieve better performance on downstream tasks such as action recognition. However, this work shows that, for the same amount of training data, AVSA improves the quality of the learned representations significantly. Due to the growing popularity of AR/VR, 360°content creation is likely to grow substantially. As the number of available 360°videos with spatial audio increases, the quality of representations learned by AVSA should improve as well.

## Acknowledgements

This work was partially funded by NSF award IIS-1924937 and NVIDIA GPU donations. We also acknowledge and thank the use of the Nautilus platform for some of the experiments discussed above.

## Broader Impact

Self-supervision reduces the need for human labeling, which is in some sense less affected by human biases. However, deep learning systems are trained from data. Thus, even self-supervised models reflect the biases in the collection process. To mitigate collection biases, we searched for 360°videos using queries translated into multiple languages. Despite these efforts, the adoption of 360°video cameras is likely not equal across different sectors of society, and thus learned representations may still reflect such discrepancies.

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
