[Supplementary Material]

# Supplementary Material: Learning Representations from Audio-Visual Spatial Alignment

## A   Implementation details

In this section, we describe in detail the implementation of the proposed AVSA pre-training as well as the semantic segmentation and action recognition downstream tasks.

### A.1   Audio-visual spatial alignment

The architecture of the video and audio encoder networks, $f_v$ and $f_a$, are shown in Table 1. The feature translation networks are described in Section 3.2 and depicted in Figure 3 of the main text. These are transformer networks of base dimension 512 and expansion ration 4. In other words, the output dimensionality of the linear transformations of parameters $W_{key}, W_{qr}, W_{val}, W_0$ and $W_2$ are 512, and that of $W_1$ is 2048. Models are pre-trained to optimize loss (7) for AVC task or (9) for AVTS and AVSA tasks. AVTS models are trained using negatives obtained from the same viewpoint but different moments in time. AVSA models are obtained using negatives obtained from the same moment in time but different viewpoints. All models were trained using the Adam optimized. Pre-training hyper-parameters are summarized in Table 2.

### A.2   Semantic segmentation

For semantic segmentation, we used a lightweight FPN segmentation head. As originally proposed, lateral connections are implemented with a $1 \times 1$ convolution that maps all feature maps into a 128 dimensional space followed by a $3 \times 3$ convolution for increased smoothing. Since the FPN head is used to perform semantic segmentation of a single frame given a video clip with multiple frames, we perform global temporal pooling of the feature maps before feeding them to the lateral connections. Semantic segmentation predictions are then computed based on the features at all levels. First, features from low-resolution layers are upsampled through a sequence of $3 \times 3$ convolutions with dilation of 2 into $56 \times 56$ resolution and added together to perform pixel-wise classification. All parameters of the FPN head are trained to minimize the softmax cross-entropy loss average across all pixels. Since we are using the output of a state-of-the-art model as ground truth, we avoid using low-confidence ground-truth labels. Thus, all pixels for which the state-of-the-art model was less than 75% confident were kept unlabeled. These low confidence regions were also ignored while computing evaluation metrics. The model was trained using the Adam optimizer with batch size 20, learning rate $1e-4$ and weight decay $5e-4$ for 10 epochs. The learning rate was decayed at epochs 5 and 8. Video clips were extracted from random viewpoints within the $360°$ video, with random angular coverage between $45°$ and $90°$ for data augmentation. Color jittering and horizontal flipping was also applied.

### A.3   Action recognition

The video encoder network was evaluated on the task of action recognition using UCF and HMDB datasets. We augmented the video encoder with a linear classification layer after the global max-pooling operation, and finetune the whole network. We used Adam optimization for 100 epochs, with batch size 28, learning rate $10^{-4}$ decayed at epochs 40, 60 and 80. Performance is reported on first train/test split originally defined for the UCF and HMDB datasets.

# B  Ablations and parametric studies

We assess different components of the proposed pre-trained mechanism through several ablation and parameteric studies shown in Table 3. All models are evaluated on AVC-Bin, AVSA-Bin, semantic segmentation, and action recognition tasks as introduced in §5.2–5.4 of main text (4 crops per video are used for AVC-Bin and AVSA-Bin). We report accuracies for the AVC-Bin, AVSA-Bin tasks using 4 viewpoints, mean IoU for the semantic segmentation task and clip level accuracy for action recognition on UCF.

**Influence of spatial audio**    To demonstrate the value of spatial audio, we train the AVSA pretext task using different inputs to the audio network: single channel mono audio, two channel stereo audio, and four channel ambisonic audio. The three versions of the audio input can be easily computed from the full ambisonics signal. The mono version of audio is generated by taking the projection of the ambisonic signal into the spherical harmonics at each viewing angle. To generate stereo, we use a standard ambisonic binauralizer that models a human listener looking at each viewing angle. To generate ambisonics, we simply rotate the original signal to align with each viewing angle. Assuming a typical ambisonics format with 4 channels, this is done by applying a 3D rotation matrix to its first-order spherical harmonic components ($X$, $Y$ and $Z$ channels), while keeping the zeroth-order component ($W$ channel) fixed.

Table 3a shows substantial improvement ($\sim 7\%$) in AVC and AVSA tasks by using full ambisonics for each crop over mono audio, suggesting that the latter may not be sufficient to encode spatial information of sound sources. Using stereo audio which retains partial spatial information also improves over mono input, but with a smaller margin. For semantic tasks (segmentation and action recognition on UCF), learning with ambisonics also proved to be more effective.

**Influence of number of viewpoints**    As more viewpoints are extracted from each sample, the difficulty of the AVSA task increases since more options are provided for matching. To investigate whether the increased difficulty correlates with the quality of the learned representation, we vary the number of viewpoints during AVSA pre-training.

Table 3b shows the AVC and AVSA performance increases monotonically as more viewpoints are used. However, these gains not always translates into better performance on semantic tasks. Semantic segmentation achieved the best performance by training to discriminate 2 or 4 viewpoints, while action recognition peaked at 4 viewpoints.

**Influence of type of negative crops**    The AVSA pretext tasks uses a combination of easy and hard (spatial) negatives: Easy negatives are clips from different video *instances*. Hard (spatial) negatives are sampled from different viewpoints, but the same moment in time. We also trained a network with hard spatio-temporal negatives, which can be sampled from any viewpoint and moment in time within the video. Table 3c shows the performance of models trained with different kinds of negatives crops. As can be seen, the combination of instance-based and spatial negatives (as used by the AVSA approach) yields better performance than using instance-based negatives alone (as used by AVC approaches). This shows the use of spatial negatives is complementary to AVC. However, the results are mixed when combining AVSA with temporal negatives (as used by AVTS approaches), producing slightly better semantic segmentations, but worse UCF performance.

**Influence of curriculum learning**    Prior work indicates that curriculum learning can benefit training by starting from easier sub-tasks and progressively increase the difficulty of the task being learned. To test this hypothesis in the AVSA context, we evaluate our network trained with and without the curriculum learning strategy (first optimizing for easy negatives, i.e. AVC, then optimizing for easy and hard negatives combined). We also compare to baselines where the model is only optimized for easy or hard negatives.

Table 3d shows that training on hard negatives directly leads to the best AVC and AVSA performance. However, the learned representations significantly overfit to the pretext task, and do not transfer well to semantic tasks, as seen by the low performance on semantic segmentation and action recognition. Using a combination of easy and hard negatives proved to be beneficial for these two downstream tasks, with the curriculum learning strategy achieving the best results.

**Influence of modeling spatial context**    We propose to use a transformer network to leverage the rich spatial context of spatial audio and 360° video while translating features across the two modalities. To assess the importance of modeling spatial context, we evaluate models trained with and without

 the transformer networks. We further vary the depth of transformer module in search of a good
trade-off between model complexity and quality of learned representations.

Table 3e shows that modeling spatial context is not required to predict whether audio and video clips originate from the same sample (achieving lower AVC accuracy). However, the ability to perform spatial alignment is significantly impacted without the transformer network, showing that it is harder to perform spatial alignment without combining information from multiple viewpoints. The lack of spatial context also impacted both semantic segmentation and action recognition on UCF. For semantic tasks, a transformer of depth $D = 2$ provided a good trade-off between model complexity and model performance.

**Video Network**

| Layer | $X_s$ | $X_t$ | $C$ | $K_s$ | $K_t$ | $S_s$ | $S_t$ |
|---|---|---|---|---|---|---|---|
| video | 112 | 8 | 3 | - | - | - | - |
| conv1 | 56 | 8 | 64 | 7 | 3 | 2 | 1 |
| block2.1 | 56 | 8 | 64 | 3 | 3 | 1 | 1 |
| | 56 | 8 | 64 | 3 | 3 | 1 | 1 |
| block2.2 | 56 | 8 | 64 | 3 | 3 | 1 | 1 |
| | 56 | 8 | 64 | 3 | 3 | 1 | 1 |
| block3.1 | 28 | 4 | 128 | 3 | 3 | 2 | 2 |
| | 28 | 4 | 128 | 3 | 3 | 1 | 1 |
| block3.2 | 28 | 4 | 128 | 3 | 3 | 1 | 1 |
| | 28 | 4 | 128 | 3 | 3 | 1 | 1 |
| block4.1 | 14 | 2 | 256 | 3 | 3 | 2 | 2 |
| | 14 | 2 | 256 | 3 | 3 | 1 | 1 |
| block4.2 | 14 | 2 | 256 | 3 | 3 | 1 | 1 |
| | 14 | 2 | 256 | 3 | 3 | 1 | 1 |
| block5.1 | 7 | 1 | 512 | 3 | 3 | 2 | 2 |
| | 7 | 1 | 512 | 3 | 3 | 1 | 1 |
| block5.2 | 7 | 1 | 512 | 3 | 3 | 1 | 1 |
| | 7 | 1 | 512 | 3 | 3 | 1 | 1 |
| max pool | 1 | 1 | 512 | 7 | 1 | 1 | 1 |

**Audio Network**

| Layer | $X_f$ | $X_t$ | $C$ | $K_f$ | $K_t$ | $S_f$ | $S_t$ |
|---|---|---|---|---|---|---|---|
| audio | 129 | 100 | N | - | - | - | - |
| conv1 | 65 | 50 | 64 | 7 | 7 | 2 | 2 |
| block2.1 | 65 | 50 | 64 | 3 | 3 | 1 | 1 |
| block2.2 | 65 | 50 | 64 | 3 | 3 | 1 | 1 |
| block3.1 | 33 | 25 | 128 | 3 | 3 | 2 | 2 |
| block3.2 | 33 | 25 | 128 | 3 | 3 | 1 | 1 |
| block4.1 | 17 | 13 | 256 | 3 | 3 | 2 | 2 |
| block4.2 | 17 | 13 | 256 | 3 | 3 | 1 | 1 |
| block5.1 | 17 | 13 | 512 | 3 | 3 | 1 | 1 |
| block5.2 | 17 | 13 | 512 | 3 | 3 | 1 | 1 |
| max pool | 1 | 1 | 512 | 17 | 13 | 1 | 1 |

**Table 1:** Architecture details of R(2+1)D video network and Conv2D audio network. The video network is based of R(2+1)D convolutions, and the audio on 2D convolutions. Both video and audio networks use ReLU activations and batch normalization at each layer. $X_s$ spatial activation size, $X_t$ temporal activation size, $X_f$ frequency activation size, $C$ number of channels, $K_s$ spatial kernel size, $K_t$ temporal kernel size, $K_f$ frequency kernel size, $S_s$ spatial stride, $S_t$ temporal stride, $S_f$ frequency stride. The input to the audio network is a $N$-channel spectrogram, where $N = 1$ for experiments with mono audio, $N = 2$ with stereo and $N = 4$ with ambisonics.

| Method | bs | nv | lr | wd | cj | hf | hfov$_{min}$ | hfov$_{max}$ | in | sn | tn | $\tau$ |
|---|---|---|---|---|---|---|---|---|---|---|---|---|
| AVC | 112 | 1 | 1e-4 | 1e-5 | ✓ | 0.5 | 25 | 90 | ✓ | | | 0.07 |
| AVTS | 28 | 4 | 1e-4 | 1e-5 | ✓ | 0.5 | 25 | 90 | ✓ | | ✓ | 0.07 |
| AVSA | 28 | 4 | 1e-4 | 1e-5 | ✓ | 0.5 | 25 | 90 | ✓ | ✓ | | 0.07 |

**Table 2:** Pre-training optimization hyper-parameters. AVSA models are initialized by the AVC model obtained at epoch 100. bs–batch size; nv–number of viewpoints; lr–learning rate; wd–weight decay; cj–color jittering; hf–horizontal flip probability; hfov$_{min}$/hfov$_{max}$–minimum/maximum horizontal field-of-view in degrees; in/sn/tn–instance/spatial/temporal negatives; $\tau$–InfoNCE temperature.

|            | AVC@4 | AVSA@4 | Segm  | UCF   |
|------------|-------|--------|-------|-------|
| Mono       | 82.39 | 62.95  | 34.21 | 64.90 |
| Stereo     | 84.47 | **71.11** | 34.54 | 64.68 |
| Ambisonics | **89.83** | 69.97 | **35.83** | **68.52** |

(a) Spatial audio format.

|              | AVC@4 | AVSA@4 | Segm  | UCF   |
|--------------|-------|--------|-------|-------|
| 1 Viewpoint  | 84.60 | 61.77  | 35.37 | 64.71 |
| 2 Viewpoints | 87.70 | 63.71  | **36.63** | 66.64 |
| 4 Viewpoints | 89.83 | 69.97  | 35.83 | **68.52** |
| 8 Viewpoints | **91.65** | **74.64** | 34.84 | 66.44 |

(b) Number of viewpoints.

|                     | AVC@4 | AVSA@4 | Segm  | UCF   |
|---------------------|-------|--------|-------|-------|
| Instance            | 83.87 | 61.20  | 34.05 | 64.09 |
| + Spatial           | **89.83** | 69.97 | 35.83 | **68.52** |
| + Spatial + Temporal| 89.65 | **72.81** | **36.11** | 65.77 |

(c) Negative crop type.

|               | AVC@4 | AVSA@4 | Segm  | UCF   |
|---------------|-------|--------|-------|-------|
| Easy Only     | 83.87 | 61.20  | 34.05 | 64.09 |
| Hard Only     | **93.22** | **77.71** | 20.97 | 59.15 |
| No Curriculum | 91.93 | 71.77  | 35.29 | 65.49 |
| Curriculum    | 89.83 | 69.97  | **35.83** | **68.52** |

(d) Curriculum learning.

|                        | AVC@4 | AVSA@4 | Segm  | UCF   |
|------------------------|-------|--------|-------|-------|
| Direct Prediction      | **91.67** | 68.87 | 34.50 | 65.59 |
| Transformer (Depth=1)  | 90.64 | **72.95** | 35.77 | 66.97 |
| Transformer (Depth=2)  | 89.83 | 69.97  | 35.83 | **68.52** |
| Transformer (Depth=4)  | 89.86 | 70.09  | **35.97** | 66.88 |

(e) Modeling spatial context.

**Table 3:** Ablation studies.