[Reviews · NeurIPS 2020]

Review 1

Summary and Contributions: This paper seeks to investigate the power of learning self-supervised audio-visual representations based on 360 degree video with spatial audio. In particular, they compare learning audio-visual spatial correspondences (AVSA) vs. the previously introduced AV tasks of either clip-level (AVC) or temporal correspondence (AVTS). Superiority of AVSA over AVC and AVTS is demonstrated on the pretext tasks themselves, as well as downstream applications.

Strengths: + The introduced self-supervised task of AV spatial correspondence is interesting and novel, and its potential of facilitating better representations based on richer information is well motivated. + The transformer-based architecture seems to be doing a good job of leveraging multiple viewpoints, and changes made to original transformer architecture make sense. + Evaluation of AVSA-based features vs AVC and AVTS is thorough, and demonstrates clear superiority of the AVSA features over the others in the presented setting.

Weaknesses: - The main weakness is in the significance of the contribution of this paper: While I am convinced that AVSA does indeed learn better features than the other tasks on this dataset, I'm not sure what the practical significance of this is. I would have liked to see novel applications and/or state-of-the-art downstream task results that are **enabled** by the proposed model. Without such objective indication of significance, we are left with only relative significance vs. previously proposed tasks. - While it does make sense that context-aware processing of multiple viewpoints is more effective, I'm uncertain of the need for the more sophisticated transformer-based modelling of the task as "translation" between video and audio. I wonder if a sufficient context-aware baseline would have been to think of the problem as a "puzzle", similar to [1,2] where the network predicts the correct arrangement of the a_i, v_j pairs by solving a supervised classification problem. [1] Doersch, Carl, Abhinav Gupta, and Alexei A. Efros. "Unsupervised visual representation learning by context prediction." ICCV 2015. [2] Kim, Dahun, Donghyeon Cho, and In So Kweon. "Self-supervised video representation learning with space-time cubic puzzles." AAAI 2019

Correctness: The paper does appear to be correct.

Clarity: The paper is well written, although one needs to read the supplement in order to learn about some important ablation studies.

Relation to Prior Work: In short: Yes, it is clear how the task and proposed model in this work differ from those in previous works. However, the significance of the proposed solution is not clear in relation to previous works: the reader does not know whether the learned features are *objectively* good for action recognition or semantic segmentation, only how good they are in relation to the provided baseline tasks. See Weaknesses section for more on this.

Reproducibility: Yes

Additional Feedback: To summarize the above: The proposed task of learning representations from 360 degree audio-visual spatial correspondences is interesting, and I believe should enable some novel and interesting applications. However, until these are shown, I think this paper's contribution is lacking in significance. UPDATE: My point was (obviously) *not* that self-supervised learning is worthless, and clearly the fact that it underperforms full supervision is not a reason to kill it. The point I *was* trying to make is that I find new SSL work interesting if (A) it pushes the boundary of SSL results on interesting/important tasks, and/or (B) it *enables* solving new tasks that were unfeasible before. While I completely agree that 360 SSL should be able to do at least one of either A or B, this paper does not show that. While you do show that on a small dataset your method is superior, I don't find the claim of "360 SSL will be SOTA in due time" to be a strong enough argument. Some (B) applications (some of which you mentioned) would be sound localization / separation / stereo or ambisonic generation, etc.


Review 2

Summary and Contributions: The authors present an approach to learn representations from 360 audio-video streams using contrastive learning. This is done in two phases: first, the model is trained to identify audio-visual correspondences at the video level and then the model is tasked with identifying the correspondence only from different crops of the video. The authors claim that this results in better features in the learned model and validate this over multiple downstream tasks (semantic segmentation and action recognition).

Strengths: 1. The new dataset based on 360 degree audio and video from Youtube would be very interesting for the community. 2. The proposed training method brings boost to a variety of downstream tasks (semantic segmentation and action recognition).

Weaknesses: 1. While the motivation of the paper is to leverage spatial information in representation learning, the model is not evaluated on an audio-visual spatial task (like localizing source of sounds of videos). The architecture also uses a max-pool making the learned model similar to previous work in literature [1]. The authors do validate on semantic segmentation task but the pixelwise labeling seems to be done purely on the basis on visual object categories that is a person is labeled even if they might be silent. The title of the paper does not seem justified as it is not clear where the "spatial alignment" is happening in the architecture or evaluation. 2. The boost in performance in semantic segmentation task is marginal over the baselines. How are the baselines able to get such good performance when localization is not involved directly in the training? How are the 4 crops chosen during training? Does the manner of selection of crops affect performance? 3. What is the performance of the method on the semantic segmentation task when a pre-trained ImageNet and randomly initialized model with a segmentation head ? 4. Are there any experiments using the Regressive AVSA? How does it compare to Contrastive AVSA? Can the rotation space be made discrete so that loss is based on classification as in [2] or use Von Mises kernel[3]? [1] Objects that Sound. Relja Arandjelović, Andrew Zisserman [2] Convolutional Neural Networks for joint object detection and pose estimation: A comparative study. Francisco Massa, Mathieu Aubry, Renaud Marlet [3] How useful is photo-realistic rendering for visual learning?Yair Movshovitz-Attias, Takeo Kanade, Yaser Sheikh

Correctness: The method is correct.

Clarity: The paper is well written.

Relation to Prior Work: Related work is discussed properly.

Reproducibility: Yes

Additional Feedback: 1. "they completely disregard the spatial cues of audio and visual signals naturally occurring in the real world." Saying the models completely disregard spatial information is too strong a statement as these models can easily be repurposed to localize sound sources to some extent. [1,2] 2. "This is accomplished by using a translation network similar the transformer of" Typo - similar to. [1] Objects that Sound. Relja Arandjelović, Andrew Zisserman [2] Learning to Localize Sound Sources in Visual Scenes: Analysis and Applications Arda Senocak, Tae-Hyun Oh, Junsik Kim, Ming-Hsuan Yang, In So Kweon. ********POST REBUTTAL******** I appreciate the authors answering my questions in the rebuttal. I believe there is some miscommunication. I meant using the model for a downstream task that requires audio visual spatial alignment. The authors report results of the AVSA self-supervision task and compare it to other methods like AVC. But that is the self-supervision task or pre-text task setup rather than an actual downstream task. The authors state themselves "Not surprisingly, AVSA performs best". "Quantitative comparison to prior work is infeasible, since localization ability has historically been shown qualitatively" I am not sure I agree with this. If a dataset with annotations of spatial sound sources is collected, the authors can show quantitatively how their method compares to other methods. The authors can use their own implementation of previous work (like AVC) to do this. However, I feel the paper has 2 strong contributions: a 360 degree audio-video dataset and a self-supervision approach for audio-visual spatial alignment. For this reason, I'll update my score. But the evaluation method focuses more on action recognition and semantic segmentation and arguably the true potential of this kind of self-supervision may lie elsewhere.


Review 3

Summary and Contributions: The paper provides a new pretext task for self-supervised representation learning using 360 video data. The authors collected a new dataset named Youtube-360 for this task. The proposed method was tested on several downstream tasks such as segmentation and action recognition. The method significantly outperformed the baseline (AVC) and shows that audiovisual spatial alignment leads to better representation learning.

Strengths: The proposed method is sound, builds on prior work of self-supervised audio-visual representation learning, and provides good results in comparison to the baseline. The authors suggest that audiovisual spatial alignment on 360 video data is very useful and introduced their model. Illustrations are carefully designed

Weaknesses: 1- While the experimental results suggest that the proposed approach is valuable for self-supervised learning on 360 video data which have spatial audio, little insights are given about why do we need to do self-supervised learning on this kind of data. In particular, 1) There are currently several large audio-video datasets such as HowTo100M and VIOLIN, 2) There is not much 360 video data on YouTube in comparison to normal data. 2- For the experimental comparisons, the authors at least should report the performance with using other self-supervised learning losses. For instance, masking features, predicting next video/audio feature, or reconstructing a feature. This will be very useful for understanding the importance of introduced loss in comparison with previous ones. 3- How the videos are divided into 10s segments? 4- It would be interesting to see how this spatial alignment works. For example, aligning an audio to the video and visualizing the corresponding visual region. 5- What's the impact of batch size on performance? batch size of 28 seems small to cover enough positive and negative samples. In this case, using MoCo loss instead of InfoNCE wouldn't help?

Correctness: The paper clearly explained the modules and losses. Therefore, it seems correct from reviewer's point of view.

Clarity: The paper is well-written and easy to follow.

Relation to Prior Work: Yes.

Reproducibility: Yes

Additional Feedback: UPDATE I checked other reviews and author's response. The authors addressed my concerns and I think the paper has enough contributions to be accepted. The dataset itself is an interesting contribution to the video-audio community. However, I still think that the authors couldn't clearly motivate why their proposed self-supervise learning framework is better than other recent works. Also, as other reviewers mentioned the evaluation of method is not complete. These are important questions, however not so critical to reject the paper. Therefore, I keep my score.


Review 4

Summary and Contributions: This paper investigates a novel self-supervised learning objective based upon predicting the spatial alignment between 360-degree audio-visual signals. For this purpose, the paper introduces a new 360-degree video dataset (YouTube-360) with Ambisonics spatial audio collected from YouTube (5.5k videos split into 88k clips, covering 246 hours of video). Similar to other contemporary works in audio-visual representation learning, a two-stream convolutional neural network model (one video stream, one audio stream) is used to produce encodings for a given pair of audio and video inputs, and the model is trained to predict whether the input pair is matched using a contrastive loss. The key algorithmic contribution of this paper is the framing of this task around predicting whether the video and audio signals are spatially aligned; previous work on self-supervised cross-modal learning from videos typically predicts whether the audio and video inputs come from the same source video, or whether they are temporally aligned (in the case that the inputs are sampled from the same source video). Experimental results are presented for the model's audio-visual spatial alignment accuracy, as well as using the learned representations for semantic segmentation (on the YouTube-360 dataset) and action recognition (on UCF and HMDB). The authors demonstrate that predicting spatial alignment outperforms predicting video-level correspondence or temporal alignment.

Strengths: Audio-visual spatial alignment is a very natural self-supervised learning objective which has eluded prior work due to the lack of appropriate datasets. This paper introduces a dataset specifically geared for this task, which is a strong contribution to the community. The paper also provides experimental confirmation that audio-visual spatial alignment is a powerful learning objective compared to temporal alignment or video-level correspondence. Finally, the paper is extremely well written and was a pleasure to read.

Weaknesses: I think that this is already a strong paper, but could be stronger if the experiments were pushed a little further. Specifically, for each potential rotation within a video clip, only a single monaural audio signal was generated from the Ambisonics sound field. It would have been straightforward to generate a stereo audio signal (or even experiment with 3 or more channels), and I expect that this would provide an even stronger learning signal for the spatial alignment prediction task. It also would have been interesting to see whether the AVSA objective is complementary with the AVC and AVTS objectives. Finally, it would have been interesting to see at least a partial confusion matrix for the segmentation task to see how the errors made by AVC/AVTS compare to AVSA. One of the motivations for AVSA stated in the introduction is that AVC/AVTS face ambiguity when confronted with frequently co-occurring objects (such as cars and roads), and that AVSA has the potential to resolve these ambiguities. It would have been nice to see verification that these kinds of errors are in fact reduced when using AVSA.

Correctness: The methodology and claims are correct.

Clarity: The paper is very well-written throughout and easy to read.

Relation to Prior Work: The paper provides a thorough overview of prior work in the area, and clearly positions itself relative to these works.

Reproducibility: Yes

Additional Feedback: Post-rebuttal feedback: In their rebuttal, the authors state: "Why using mono? Audio input ablation (R4) There is a misunderstanding here. We use full ambisonics aligned with each viewpoint by a 3D rotation (L195-196), and compare to mono/stereo inputs in suppl. Table 3a" I think the author(s) have misunderstood my question here. I understand that the full ambisonics are used to produce an audio waveform aligned with each viewpoint. However, the resulting aligned audio still appears to be a mono signal (meaning a single spectrogram is computed for each rotation), according to Table 1 in the supplementary material. This is analogous to attaching a single, forward-facing directional microphone to the front of the camera. The authors are effectively using the ambisonics to perform beamforming, attenuating the volume of sounds originating from outside the current FOV while boosting the volume of sounds in the current FOV. However, for a given FOV, that single spectrogram doesn't preserve direction of arrival information for the different sound sources that are present. The proposed approach is actually under-utilizing the ambisonics; I imagine the method would be even more powerful if a stereo audio signal (or even using a greater number of channels) was computed for each rotation. I am inclined to keep my originally assessed score. I really appreciate the novelty of this paper and think that using 360 degree audio-video data is going to be important in the future. I agree with the other reviewers that a more thorough exploration of novel tasks enabled by this approach and/or demonstration of improvements over the current SotA on downstream tasks would have made this paper much stronger, but given the novelty here I don't think the lack of those things is reason enough to reject this paper.

[Author Response · NeurIPS 2020]



Figure 1: Sound localization maps from AVC (**1st** of each pair) and AVSA (**2nd** of each pair) models.

We thank reviewers for the constructive feedback. We're glad reviewers found AVSA novel, well-motivated (R1,R4),
and superior to prior work in several tasks (R1,R2,R3,R4); and appreciated our dataset (R2,R4). We'll address minor
points in the paper. Some requested ablations were already in suppl. Table 3. We'll summarize them in the paper.

**Significance (R1) / Why 360 videos? (R3)** We respectfully disagree with R1. The standard of "I see no practical
significance" could be used to kill most SSL research, as it rarely outperforms full supervision. Applying this standard
to 360°SSL seems equally short sighted. It is true that 360°SSL currently underperforms other forms of SSL for
tasks like action recognition. This is because 360°datasets are much smaller. However, with the push for AR/VR and
self-driving cars, this is temporary. The 360°camera market is projected to grow 25.6% annually until 2027 [link], and
content availability should grow even faster. Furthermore, 360°SSL will enable new applications. AVSA can be used, as
is, to detect and correct misalignments between 360°cameras and external microphones commonly used when shooting
high quality 360°content. AVSA should also enable better audio localization, since it provides direct spatial supervision
by matching co-located audio and video crops. However, no labeled video datasets exist for evaluation (only a small set
of static images [50]). While we aim to annotate a sound localization dataset, this is beyond the scope of this work.
In conclusion, there is a flourishing literature on audio-visual SSL methods, which are SOTA for SSL in tasks like
action recognition, and more will appear at NeurIPS. We show that, on the same dataset, AVSA clearly beats these
methods, even on the AVC task. Hence, there is no reason to believe that 360°SSL will not be SOTA in due time.

**Alternative implementations: Context-aware puzzles (R1) / Regressive AVSA (R2) / Other SSL losses (R3).**
These are great suggestions, although they shouldn't challenge the main contribution (showing that spatial information
boosts representation learning). Given the short rebuttal cycle, we didn't try all, but will add a full comparison to the
camera ready. One advantage of our method is the seamless combination of audio-visual correspondence (AVC) and
spatial alignment (SA) by simply extending the negative set. Suppl. Table 3c shows that this combination is crucial. A
puzzle-like approach (R1), where the arrangement of audio/video crops is predicted *simultaneously* by an MLP, only
handles SA and thus performs worse (61.7% video@1 on UCF vs. 73.8% for AVSA). Combining with AVC improves
performance (70.3%), but is still less effective. We tried Regressive AVSA (R2), but it performed much worse as well
(66.7%). Per R3 suggestion, we experimented with masking random $K$ audio/video crops and using the masked features
to setup a multiple choice problem. The best design (K=4) was worse than AVSA (63.3% w/o AVC, 70.4% w/ AVC),
likely due to unmasked video crops making SA of other video crops easier. We didn't try single modality SSL losses,
eg next frame prediction, as prior work [1,3,30,41] show the superiority of cross-modal over within-modal supervision.

**"Spatial alignment" in title (R2)** There is a misunderstanding. Spatial alignment is our main source of information
for representation learning (L154-161), as we seek to match audio and video crops *from the same location*. And we do
*evaluate* on audio-visual spatial alignment directly (Table 2, AVSA-Bin task). Not surprisingly, AVSA performs best.

**Evaluation by sound source localization (R2) / visualizing alignment (R3)** Fig. 1 shows the GradCAMs of audio-
visual alignment scores for AVC and AVSA models. AVSA localization is more accurate. We'll add these to the paper.
Quantitative comparison to prior work is infeasible, since localization ability has historically been shown qualitatively.

**Semantic segmentation (R2)** We disagree that boost is marginal. We emphasize that only AVSA can learn context
features, which yield a 3% mIOU boost over AVC and 1% over AVTS. Comparisons to ImageNet pre-training are not
informative, since segmentation targets are from an FPN model *pre-trained on ImageNet* and tuned on COCO. Kinetics
pre-training is slightly better than AVSA (75.5% acc and 36.9% mIoU using video alone) but relies on human labels.
Random initialization performs poorly (29.7% acc and 8.5% mIoU) since we evaluate frozen representations (common
SSL practice). Full training is shown in last row of Table 3, but not comparable to frozen representations (other rows).

**Crop selection (R2)** Crops are sampled randomly, with a minimum distance enforced (L186). More sophisticated
crop selection is an interesting direction, eg guiding selection toward objects using objectness scores, moving objects
using optical flow, or sound sources using audio energy from different viewpoints. We'll add future work to the paper.

**Impact of batch size (R3)** Yes, large batches should improve AVSA. This would require (1) a lager GPU cluster for
distributed training [9] or (2) maintaining a memory bank (NPID [61]) or buffer (MoCo [23]) of features. However,
unlike [61,23], our features vary with location, and storing several features per video (one per location) is infeasible.

**Why using mono? Audio input ablation (R4)** There is a misunderstanding here. We use full ambisonics aligned
with each viewpoint by a 3D rotation (L195-196), and compare to mono/stereo inputs in suppl. Table 3a.

**Is AVSA complementary to AVC/AVTS? (R4)** Suppl. Table 3c shows that AVSA is complementary to AVC. New
experiments shows the same for AVTS only on segmentation task. Spatial negatives alone (AVSA) yield 35.8% / 68.5%
on segmentation / UCF. Spatio-temporal negatives (ie AVSA+AVTS) yield 36.1% / 65.8%. We'll add to the paper.

**Understanding AVSA gains (R4)** We compared class accuracies of AVSA and AVC on segmentation. AVSA indeed
improves the most on background classes: rocks (27.7→34.7), window (41.2→46.0), pavement (33.3→36.8), sand
(38.8→42.1), sea (46.8→50.1) and road (45.1→47.1). This suggests that AVSA disentangles common backgrounds
more effectively. We'll add a complete error analysis to the paper. Thanks for the suggestion!

[Meta-Review · NeurIPS 2020]

The paper received mixed reviews. The reviewers found 360 audio to be very interesting for self-supervised representation learning. However, at the same time, the reviewers noted that the evaluation did not align well with spatial tasks, which is where intuitively the benefits of 360 audio would transpire. The rebuttal seemed to further misunderstand several of the points raised by the reviewers (see multiple individual reviews). In discussion, reviewers were also quite puzzled by the rebuttal, especially because it misrepresented the major points they mentioned. However, although the rebuttal left much to be desired by the reviewers and AC, the reviewers generally agreed the points were not sufficient to reject the paper. Given the other contributions of the paper as noted by the reviewers, the AC recommends the paper for acceptance.